# Overlap of Peak Growth Activity and Peak IGF-1 to IGFBP Ratio: Delayed Increase of IGFBPs Versus IGF-1 in Serum as a Mechanism to Speed up and down Postnatal Weight Gain in Mice

**DOI:** 10.3390/cells9061516

**Published:** 2020-06-22

**Authors:** Michael Walz, Luong Chau, Christina Walz, Mandy Sawitzky, Daniela Ohde, Julia Brenmoehl, Armin Tuchscherer, Martina Langhammer, Friedrich Metzger, Christine Höflich, Andreas Hoeflich

**Affiliations:** 1Institute of Genome Biology, Leibniz-Institute for Farm Animal Biology (FBN), 18196 Dummerstorf, Germany; walz.michael@fbn-dummerstorf.de (M.W.); chau@fbn-dummerstorf.de (L.C.); walz@fbn-dummerstorf.de (C.W.); mandysawitzky@gmx.de (M.S.); ohde@fbn-dummerstorf.de (D.O.); brenmoehl@fbn-dummerstorf.de (J.B.); 2 Institute of Genetics and Biometry, Leibniz-Institute for Farm Animal Biology (FBN), 18197 Dummerstorf, Germany; atuchsch@fbn-dummerstorf.de (A.T.); martina.langhammer@fbn-dummerstorf.de (M.L.); 3 Versameb AG, 4057 Basel, Switzerland; friedrich.metzger@versameb.com; 4Ligandis UG, 18276 Gülzow-Prüzen, Germany; christine.hoeflich@ligandis.de

**Keywords:** longitudinal study, IGFBP, mouse models

## Abstract

Forced expression of insulin-like growth factor binding proteins (IGFBPs) in transgenic mice has clearly revealed inhibitory effects on somatic growth. However, by this approach, it cannot be solved if or how IGFBPs rule insulin-like growth factor (IGF)-dependent growth under normal conditions. In order to address this question, we have used growth-selected mouse models (obese and lean) and studied IGF-1 and IGFBPs in serum with respect to longitudinal growth activity in males and females compared with unselected controls. In mice of both genders, body weights were recorded and daily weight gains were calculated. Between 2 and 54 weeks of age, serum IGF-1 was determined by ELISA and intact IGFBP-2, -3 and -4 were quantified by Western ligand blotting. The molar ratio of IGF-1 to the sum of IGFBP-2 to -4 was calculated for all groups and plotted against the daily weight gain curve. Growth-selected mice are characterized by higher daily weight gains and extended periods of elevated growth activity if compared to matched unselected controls. Therefore, adult mice from the obese and lean groups can achieve more than twofold increased body weight in both genders (*p* < 0.001). Between 2 and 11 weeks of age, in obese and lean mice of both genders, serum IGF-1 concentrations are increased more prominently if compared to unselected controls (*p* < 0.001). Instead, substantial decreases of IGFBPs, particularly of IGFBP-2, are observed in males and females of all groups at the age of 2 to 4 weeks (*p* < 0.001). Due to the strong increase of IGF-1 but not of IGFBPs between two and four weeks of age, the ratio of IGF-1 to IGFBP-2 to -4 in serum significantly increased in all groups and genders (*p* < 0.05). Notably, the IGF-1 to IGFBP ratio was higher in male and female obese mice if compared to unselected controls (*p* < 0.05).

## 1. Introduction

Long-term selection for high body weight goes back to 1930, when Goodale initiated an experiment to explore the boundaries of growth in mice [1]. After 35 generations of selection, the mice had increased their body weight from ≈25 g to ≈43 g (+72%). Most probably due to inbreeding effects, additional selection for 49 generations did not further increase body weight to a significant extent [2]. Starting from an outbred background and under avoidance of inbreeding in the present selection experiment, substantial increases (+144%) of male body weight at the age of six weeks were achieved after 146 generations of selection in the obese mouse line (DU6) [3]. This finding not only underlines the potential of non-inbred backgrounds for functional genome analysis but even more importantly proves the idea that growth is a complex trait regulated by a multitude of effectors [4]. Here we have used two separate growth selected mouse models for the study of longitudinal regulation of the IGF-system. Accordingly, mice long-term selected for high body mass [3,5,6] (obese model; DU6), and a second mouse model selected for high protein mass [7,8] (lean model; DU6P) were compared to unselected controls [9,10].

Clearly, the GH–IGF system is highly responsive to growth selection; specific effects have been described on the level of DNA, mRNA, and protein with respect to the GH–IGF system in model animals or farm animals [11,12,13,14,15], and many of these studies have particularly addressed the biomarker potential of IGF-1 or assessed single time points. In human subjects, longitudinal concentrations have been provided both for IGF-1 [16] and for IGFBP-3 [17]. In order to estimate the bioactivity of IGF-1, reference levels for the ratio of IGF-1 to IGFBP-3 were also calculated for male and female subjects from a larger population longitudinally [17]. In biological matrices and in the circulation, IGF-1 bioactivity is not only regulated by IGFBP-3, and therefore the inclusion of additional IGFBPs enables a more comprehensive view e.g., on the control of IGF-1 dependent growth. For the hypothesis-free assessment of IGFBPs in a given matrix, Western ligand blotting (WLB) technique can be applied [18]. By this method, it is possible to include all IGFBPs present and detectable in a given sample. Perhaps even more important is the fact that WLB delivers structural information of a given IGFBP [19]. Thereby the information provided by WLB is related to a specific molecular weight (e.g., intact IGFBP-3), whereas other methods do not have this power. This fundamental feature of WLB is getting more and more important, as we understand that IGFBP-proteolysis represents a fundamental process of physiological growth control related to IGFs [20,21] or in cancer [22]. Just recently, an IGFBP-3 protease has been described as an effector of free IGF-1 in children and adolescents [23]. Accordingly, the inclusion of structural information could tremendously improve the biomarker value of IGFBPs [19]. Here we compared intact IGFBPs quantified by WLB with longitudinal concentrations of IGF-1 and, for the first time, discuss IGF-1 to IGFBP ratio based on structurally validated biomarker information of IGFB-2, -3 and -4 in serum.

## 2. Materials and Methods

### 2.1. Animals, Husbandry, and Study Design

In the present study, long-term selected non-inbred mouse lines established at the Leibniz Institute for Farm Animal Biology (FBN) were used. Two lines were selected for high male body mass at the age of 42 days (DU6: obese model) or high protein amount (DU6P: lean model) at the same age of 42 days after birth. These long-term selected mouse lines were originally based on the genetic background of the unselected control line Fzt:DU [9,10]. The control mouse line (Fzt:DU) was developed by random mating procedures during the experiment. Husbandry, mode of selection, and phenotypical features of the three mouse lines have been described in detail before [3,8]. In brief, all mice were maintained under semi-barrier conditions with free access to chow (breeding diet 1314, Altromin, Lage, Germany) and water. In order to assess longitudinal levels of IGFBPs in serum from male and female mice, we used serum produced by Sawitzky et al. [8]. In the course of this study, male and female mice from all three lines were dissected at the age of 2, 4, 7, 11, 16, 29, 42, and 54 weeks after birth, and serum was frozen until further use. The experiment was designed with 8 animals per group. Due to elevated mortality, only 4 male obese mice reached an age of 54 weeks, resulting in a total sample number of N = 380. In addition, body weights were recorded from all mice included in this study. Daily weight gain was calculated from intrapolated daily weights extracted from the Gompertz growth curve (Y = YM*(Y0/YM)^(exp(−K*X))). The experiments were performed in adherence to national and international laws and were further approved by the National Animal Protection Board Mecklenburg-Vorpommern (file number: LALLF M-V/TSD/7221.3-1.2-037/06).

### 2.2. Longitudinal Analysis of IGFs and IGFBPs in Mouse Serum

In serum from male (N = 188) and female (N = 192) mice between 2 and 54 weeks of age, IGF-1 was quantified by ELISA as described before [24]. In all samples, IGFBPs were quantified by Western ligand blotting as already described [25] with exceptions as described here. Serum was denatured for 5 min at 95 °C in sample buffer containing 10% sucrose, 2% sodium dodecyl sulfate (SDS), and 62.5 mM Tris (pH 6.8) and loaded on 12%-SDS/polyacrylamide gels. For quantification of IGFBP-2, -3, and -4, dilution series of human recombinant IGFBP-2, -3, and -4 were included with each run. After electrophoresis, proteins were blotted from the gel to solid carrier membranes (polyvinyl fluoride, Millipore, Schwalbach, Germany). The membranes were incubated using human recombinant IGF-2 radiolabeled with iodine-125 overnight at 4 °C. After five consecutive repetitions of washing in phosphate-buffered saline (pH 7.4) for 15 min, membranes were exposed to Storage Phosphorimager screens mounted on plates for 8 h. The signals were quantified using the Phosphor-Imager Storm (Molecular Dynamics, Sunnyvale, CA, USA). Quantification was achieved using ImageQuant software (GE Healthcare, Marlborough, MA, USA). Regression coefficients from standard dilutions were higher than 0.99 (http://www.ligandis.de/index.php?id=20&L=1). Intraassay and interassay variations were <15% and <20% for all IGFBPs, as published before [25]. Lower limits of quantification also as published before [25] were 0.25 ng for IGFBP-2 and 1 ng for IGFBP-3 and IGFBP-4. Using the software GraphPad Prism version 8.4.2, three samples were identified as outliers (GraphPad Prism) and therefore excluded from further analysis. Accordingly, in male controls at an age of 2 weeks, in obese males at an age of 42 weeks, and in female controls at an age of 42, only 7 samples per group were included (N = 377).

### 2.3. Statistical Analyses

Statistical analyses were performed using the SAS software for Windows, version 9.4 (Copyright, SAS Institute Inc., Cary, NC, USA). IGF-1 and IGFBP and growth data were analyzed by analyses of variance (ANOVA) using the MIXED procedure in SAS/STAT software. The ANOVA model contained the fixed factors group (levels: obese, lean, control), gender (levels: female, male), age (levels: weeks 2, 4, 7, 11, 16, 29, 42, 54), and their interactions.

Least square means (LS means) and their standard errors (SE) were computed for each fixed effect in the models described above, and all pairwise differences between LS means were tested using the Tukey–Kramer procedure. The SLICE Statement of the MIXED procedure was used to perform partitioned analyses of the LSM for all interactions. Effects and differences with *p*-values < 0.05 were considered significant.

## 3. Results

### 3.1. Longitudinal Growth in Non-Inbred Mouse Models

Body weight was recorded in mice selected for high body mass at the age of 42 days (obese mouse model), in mice selected for high protein amount (lean mouse model), and in unselected controls (Figure 1). At an age of 11 weeks in females and 16 weeks in males, body weights were significantly different between obese, lean, and control mice. Within groups, daily weight gains were highest in male or female controls at the age of 24.9 days or 20.9 days after birth. Lean mice elevated daily weight increases until an age of 26.3 days in males and 24.5 days in females. In obese mice, the daily weight gains peaked at the age of 26.7 days in male and 24.2 days in female mice. The absolute amount of daily weight increase amounted to 0.635 g/d and 0.59 g/d in male and female unselected controls, 1.99 g/d and 1.52 g/d in lean male and female mice, but 2.2 g/d and 1.7 g/d in obese male and female mice.

### 3.2. Effects of Age and Growth Selection on the Concentrations of IGF-1

As a main effect of growth selection independent of age, IGF-1 was significantly increased in lean mice (*p* < 0.001) of both genders but only in obese male mice (*p* < 0.01) if compared to sex-matched unselected controls (Figure 2). As an effect of age and genetic group, in male obese mice, a significant (*p* < 0.001) increase of IGF-1 concentrations in serum was present between 2 and 4 weeks of age. In male lean mice, a similar increase was found between 2 and 7 weeks of age (*p* < 0.001). In males from both growth-selected mouse lines, IGF-1 concentrations decreased between 4 or 7 and 29 weeks of age (*p* < 0.01). Moreover, as an interaction of age and genetic group, in female obese and lean mice, increases of IGF-1 concentrations were found between 2 and 7 weeks of age (*p* < 0.001). However, a significant decrease of IGF-1 concentrations over time was only found in obese female mice between 7 and 54 weeks of age (*p* < 0.001).

### 3.3. Effects of Age and Gender on Levels of IGFBP2- to 4 

By direct comparison of longitudinal IGFBP profiles in serum from male and female unselected non-inbred mice (data not shown), IGFBP-2, -3, and -4 exhibited gender-related features: if compared to age-matched females, male mice had higher concentrations of IGFBP-2 (*p* < 0.01) and -3 (*p* < 0.001) at an age of 16 and 42 weeks, respectively, but lower concentrations of IGFBP-4 in serum (*p* < 0.001) at the age of 54 weeks.

#### 3.3.1. IGFBP-2

As a main effect of age in all female mice, IGFBP-2 was reduced between weeks 2 and 4 (*p* < 0.01), increased between weeks 4 and 11 (*p* < 0.001), and reduced between weeks 11 and 16 (*p* < 0.001; Figure 3). As an effect of age in all male mice, IGFBP-2 also was reduced between weeks 2 and 4 (*p* < 0.001), increased from week 4 until week 16 (*p* < 0.001), and then decreased from week 16 to week 26 (*p* < 0.001). The effects of age in selected mouse lines are depicted in Figure 3 and Figure 4 (interactions of age and genetic group).

#### 3.3.2. IGFBP-3

As an effect of age in all female mice, a substantial increase of IGFBP-3 (Figure 4) in serum was observed between 2 and 16 weeks of age (*p* < 0.001). In addition, a reduction was observed in all female mice between week 16 and week 54 (*p* < 0.001). Similarly in all male mice, an increase of IGFBP-3 in serum between weeks 2 and 11 (*p* < 0.001) and a decrease between weeks 11 and 54 (*p* < 0.001) was observed. Significant effects of age are presented for separate mouse lines in Figure 3 and Figure 4 (interactions of age, gender, and mouse line).

#### 3.3.3. IGFBP-4

As an effect of age, IGFBP-4 was significantly decreased between weeks 2 and 7 (*p* < 0.05), increased between week 7 and 11 (*p* < 0.05) and decreased between week 11 and 42 (*p* < 0.001). In all male mice, IGFBP-4 was increased between weeks 4 and 11 (*p* < 0.01) and decreased between weeks 11 and 29 (*p* < 0.001). Again, significant differences present in isolated mouse lines are depicted in Figure 3 and Figure 4.

### 3.4. Effects of Growth Selection on the Concentrations of IGFBP-2 to -4

As a main effect of growth selection and irrespective of age, obese female mice had significantly higher levels of IGFBP-3 in serum (*p* < 0.001). Independent of age, selection for high protein accretion increased IGFBP-3 and IGFBP-2 (*p* < 0.001). In males, selection for high body mass and selection for high protein mass had an effect on the concentrations of IGFBP-2 and IGFBP-3 in serum (*p* < 0.05) over all age groups.

As an interaction of genetic group and age, in growth-selected obese and lean male mice (Figure 4), levels of IGFBP-3 at the age of 11 and 16 weeks were increased if compared to age-matched unselected controls (*p* < 0.05). In lean male mice, IGFBP-3 remained on a higher level also at later time points, with significant differences if compared to obese male mice and unselected controls at the age of 29 and 54 weeks (*p* < 0.05). Growth selection further stimulated the increase of IGFBP-2 from younger ages to week 16, observed in unselected controls, resulting in about 3-fold increased levels of IGFBP-2 in serum from lean male mice (*p* < 0.001). Between 4 and 11 weeks of age, a significant increase of IGFBP-4 was observed (*p* < 0.01) in all males independent of line. In all genetic groups and in both genders, the postnatal increase of IGFBP-3 in serum is lagging behind the increases of IGF-1.

### 3.5. Longitudinal Molar Ratio of IGF-1 and IGFBP Concentrations in Serum

In order to estimate the longitudinal molar ratio of IGF-1 with respect to the IGFBPs detected in serum by Western ligand blotting, the concentrations of IGF-1 and IGFBP-2 to -4 were corrected for their respective molecular weights (IGF-1: 7.5 kDa, IGFBP-2: 32 kDa, IGFBP-3: 41 kDa, IGFBP-4: 24 kDa). The longitudinal molar ratio of IGF-1 versus the sum of IGFBP-2, IGFBP-3, and IGFBP-4 is presented in Figure 5. Neither in female nor in male mice of all genetic groups, IGF-1 was in molar excess over the sum of IGFBP-2 to -4. As an effect of age and genetic group, between weeks 2 and 4, there was a significant increase in all groups and genders (*p* < 0.001) with the exception of female controls (*p* < 0.05). At the age of 4 weeks, obese male and female mice have higher ratios of IGF-1/IGFBPs if compared to unselected controls (*p* < 0.05).

## 4. Discussion

Functional genome analysis in genotype-based (transgenic or knockout) mouse models has identified multiple functions of the IGF system for somatic growth [26,27,28]. By contrast, descriptive studies using phenotype-derived mouse models have been used to a much lesser extent. Nevertheless, phenotype-derived models can provide important information on growth regulation under physiological conditions and may also be useful for the identification and validation of biomarkers. 

For the establishment of those models, phenotype selection in mice was initiated in 1976 in Dummerstorf (Germany) based on a mixed genetic background comprising four different inbred and four different outbred mouse models [10]. Growth selection has been performed with respect to male body weight at the age of 42 days, resulting in an obese model characterized by extreme body weight and marked obesity resulting in the DU6 mouse model [5]. In addition, a lean mouse model (DU6P) has been developed from the identical genetic background by selection for high protein amount in the whole body [29]. Line-specific accumulation of body fat and the accretion of muscle mass over time is described elsewhere [3,8]. Here we describe endocrine parameters of IGF-related growth in male and female mice from obese and lean mice, compared to unselected controls, in a longitudinal setting. For the analysis, we included exclusively intact IGFBPs (IGFBP-2 to -4) detectable in serum by Western ligand blotting.

Line-, gender- and age-specific growth characteristics were identified by the statistical model with significantly higher body mass in obese versus lean mice. Higher body mass in growth-selected mice is reflected by higher daily weight gains and prolonged pubertal growth if compared to unselected controls. 

As published before [29] and confirmed here, after birth, obese and lean mice are characterized by substantial increases of serum IGF-1 concentrations if compared to controls, which might nicely explain higher growth activity in both mouse models. However, IGF-1 and GH have common and independent effects during early postnatal growth [30]. Therefore, we have to consider the IGF-independent effects of growth hormone during the earlier postnatal growth period but also the effects of IGF-2. The potential effects of the embryonal growth factor IGF-2 [31], which also can have positive effects on body weight after birth [32], need to be studied in a separate study. Although daily weight gain declined between 17 and 27.5 days of age in all mice included in that study, the concentrations of IGF-1 in the circulation remained elevated until an age of at least 7 weeks. The elevated levels of IGF-1 therefore cannot explain the massive reductions of growth activity in mice between 4 and 7 weeks of age. In female but not male human subjects, IGF-1 concentrations also lag behind the peak weight gain in males and females [16,33]. Accordingly, highest IGF-1 serum concentrations were found at the age of about 14.6 years in male and female human subjects [17], whereas peak weight gain was referred to an age of 12 years in females and 14 years in males [34]. The clear decrease of serum IGF-1 concentrations, at least in male mice between week four and week 26 of age, identifies peak-like kinetic for serum IGF-1 concentrations as also found in human subjects [16]. In female mice, a decrease of serum IGF-1 concentrations is less clear and was only found in obese mice. In unselected controls, a peak-like pattern is virtually absent. This is a clear difference if compared to human subjects, where serum IGF-1 concentrations were clearly lower in adults compared to younger subjects [16]. 

In order to understand why daily weight gain was reversed in the presence of high or elevated IGF-1 serum concentrations, we studied serum IGFBP concentrations in all samples. In all male genetic groups, IGFBP-2 concentrations were reduced directly after birth between two and four weeks of age (effect of age by gender for males: *p* < 0.001). The reductions in male mice between two and four weeks were characterized by outmost uniformity as the curves were overlaying each other, and the standard deviations were comparably low. Another clear feature of longitudinal IGFBP-expression was seen in strong increases of IGFBP-3 in male and female mice between four and 11 weeks of age. In human subjects of both genders, IGFBP-3 increased during the growth period until the age of 20 years [17,35]. 

In lean and obese male mice, serum IGFBP-2 concentrations were elevated at week 16 after birth compared to earlier time points. In female mice, a postnatal increase occurred one month earlier at week 11 and was significant compared to two, four, and seven weeks of age, independent of the genetic group (data not shown). Due to gender-specific patterns of longitudinal IGFBP-2 concentrations in serum, females from all genetic groups had significantly lower levels of IGFBP-2 compared with their male littermates at the age of 16 weeks. In human serum, concentrations of IGFBP-2 are one order of magnitude lower compared to IGFBP-3 and decrease from childhood to adolescence [36], similar to mice. With advanced age in humans, concentrations of IGFBP-2 in serum increase, with the highest levels found during senescence [37,38]. Accordingly, we may have similarities between mice and humans only during the initial postnatal growth period.

Similar to IGFBP-2, serum levels of IGFBP-4 were reduced between weeks 2 and 4 in male mice but significantly increased in both genders between week 4 and 16 independent of group (*p* < 0.001). In healthy human subjects, serum IGFBP-4 concentrations did not change with age [39]. 

The molar ratio of IGF-1 to the molar sum of all IGFBPs identified in serum by Western ligand blotting was characterized by significant increases between week 2 and week 4 in all groups. Thereby, the IGF/IGFBP ratios reached their lifetime maxima when growth activity was also high in mice. In obese mice, at the age of four weeks, the IGF-1/IGFBP ratio was also significantly increased compared to unselected controls, which may explain at least part of the higher growth activity under conditions of growth selection. The overlay of IGF-1/IGFBP ratios with the daily weight increases are only partial in unselected controls of both genders. This may indicate that the extended periods of growth activity in growth-selected mice may be related to elevated IGF-1 and/or elevated IGF-1 bioactivity. As provided by data from larger groups of mice (data not shown), growth-selected mice are heavier already at the time of birth, where we could not identify higher IGF-1 or IGF-1/IGFBP ratios. From work in genotype-derived mouse models, we know that in particular, IGF-II and GH or other hormones like insulin have an effect on early growth and development [30]. As also mentioned earlier, the potential roles of these hormones have to be addressed in future studies. In all groups and genders, the kinetics of serum IGFBP-3 concentrations are lagging behind those of serum IGF-1 concentrations. Thereby, a mechanism may be generated for the establishment of acutely high IGF-1/IGFBP ratios. By contrast, a delayed increase of IGFBP-3 versus IGF-1 was definitely not described in humans. According to published reference levels [17], male and female children at the age of about five years already had an increase of 75% of their maximal IGFBP-3 during adulthood. Thus, a delayed increase of IGFBP-3 versus IGF-1 was definitely not described in humans [16,17], which could be due to species differences or the different analytical systems used. 

To date, it is unclear how such a shift is established in mice. In general, the altered expression of IGFBPs on the level of RNA and/or protein or altered stability of IGFBPs in biological matrices may be causative of altered concentrations of IGFBPs and thus altered bioactivity of IGF-1. An involvement of IGFBP-proteolysis for the control of height attainment has been suggested by Marouli et al. [4]. In this study, an allele of stanniocalcin 2 was characterized, which was less efficient in blocking proteolytic activity of PAPP-A. Just recently, it was demonstrated that PAPPA-2, which represents a candidate gene for growth regulation in mice as well [40], is significantly decreased during childhood and negatively correlated with intact IGFBP-3 in humans [23].

However, this study has distinct limitations. First of all, due to the longitudinal and initially descriptive approach, the number in every single age group was comparably small and further reduced by higher mortality with age, particularly in male obese mice. Accordingly, the highest age group with 4 replicates in obese males can be considered less reliable. In the future, higher sample numbers should be chosen in selected groups also for confirmatory studies. Furthermore, IGFBP-3 is present in a ternary complex, whereas IGFBP-2 and IGFBP-4 are present in binary complexes only. Therefore, differential pharmacokinetic properties can be expected for the different types of complexes. Therefore, the present study followed a simplistic approach by combining IGFBP-2 to -4 for the estimation of IGF-1 to IGFBP ratios. Future studies also would have to consider the concentrations of IGFBP-1, -5, and -6, which were not detected by Western ligand blotting due to lower sensitivity compared e.g., to ELISA. Furthermore, the different compounds from the GH/IGF-system are inter-related, as GH and IGF-1 are particular determinants of IGFBP concentrations. Finally, the physiological relevance of IGF to IGFBP ratios less than 1 needs to be addressed in future studies. This could be achieved by the analysis of free IGF-1 or IGF-related bioactivity in animals characterized by different IGF to IGFBP ratios. 

To summarize, we have characterized longitudinal concentrations of IGF-1 and intact IGFBP-2 to -4 in serum from two different mouse lines selected for high growth and unselected controls in both genders. We compared the IGF-1/IGFBP ratios with daily weight gain and were able to provide evidence that part of the elevated growth activity during prepubertal growth in normal and growth-selected mice could be related to elevated bioactivity of IGF-1. Elevated ratios of IGF-1/IGFBPs are established by a delayed increase of IGFBPs compared to strong increases of IGF-1 between 2 and 4 weeks of age.

We therefore may be in a position to distinguish two phases of IGF-1 related growth during early postnatal development: acceleration of postnatal growth by elevated serum IGF-1 concentrations followed by a phase of deceleration due to the delayed increase of IGFBPs in serum.

## Figures and Tables

**Figure 1 cells-09-01516-f001:**
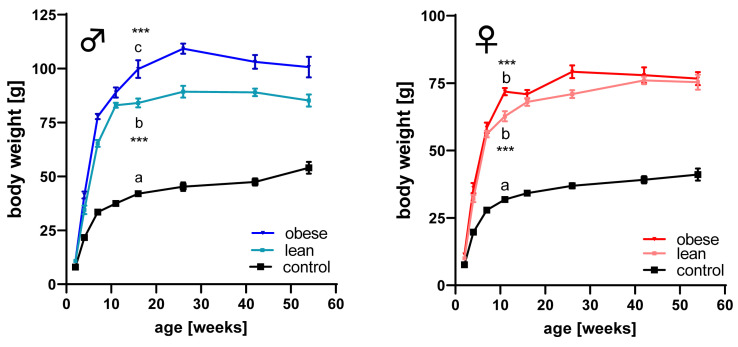
Body weight in male (left panel) and female (right panel) mice selected for high body weight (obese), for high protein amount (lean), and in unselected controls at an age of 2, 4, 7, 11, 16, 29, 42, and 54 weeks. (mean ± SEM; *n* = 8; due to high mortality, sample number was reduced to *n* = 4 at an age of 54 weeks in male obese mice; different letters (a, b, and c) indicate significant differences also in different genetic groups per gender, ***: *p* < 0.001; identical letters indicate no statistically significant difference).

**Figure 2 cells-09-01516-f002:**
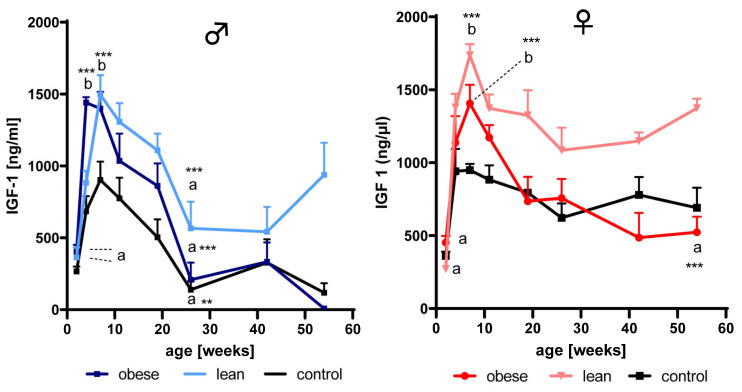
Concentrations of insulin-like growth factor (IGF)-1 in serum from mice selected for high body weight (obese), for high protein amount (lean), and in unselected controls of both genders at an age of 2, 4, 7, 11, 16, 29, 42, and 54 weeks. Different letters (a and b) indicate significant differences also with respect to different genetic groups in each gender; identical letters indicate no statistically significant difference. Sample information is provided by Figure 1 (mean ± SEM; **: *p* < 0.01; ***: *p* < 0.001).

**Figure 3 cells-09-01516-f003:**
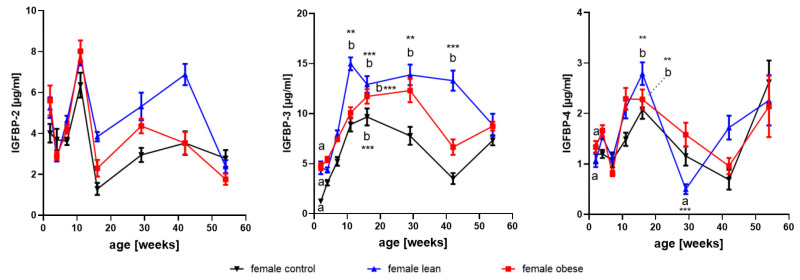
Concentrations of insulin-like growth factor binding protein (IGFBP) -2, -3, and -4 in serum from female mice selected for high body weight (obese), for high protein amount (lean), and in unselected controls at an age of 2, 4, 7, 11, 16, 29, 42, and 54 weeks (mean ± SEM; n ≥ 7; different letters (a and b) indicate significant effects also if different genetic groups were compared; identical letters indicate no statistically significant difference; **: *p* < 0.01; ***: *p* < 0.001).

**Figure 4 cells-09-01516-f004:**
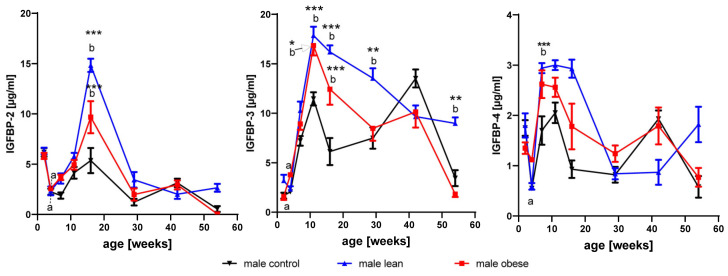
Concentrations of IGFBP-2, -3, and -4 in serum from male mice selected for high body weight (obese), for high protein amount (lean), and in unselected controls at an age of 2, 4, 7, 11, 16, 29, 42, and 54 weeks (mean ± SEM; *n* = 8 with the exception of obese male at an of 42 weeks and 54 weeks with *n* = 7 and *n* = 4, respectively; different letters (a and b) indicate significant effects of age; identical letters indicate no statistically significant difference also if different genetic groups were compared; *: *p* < 0.05; **: *p* < 0.01; ***: *p* < 0.001).

**Figure 5 cells-09-01516-f005:**
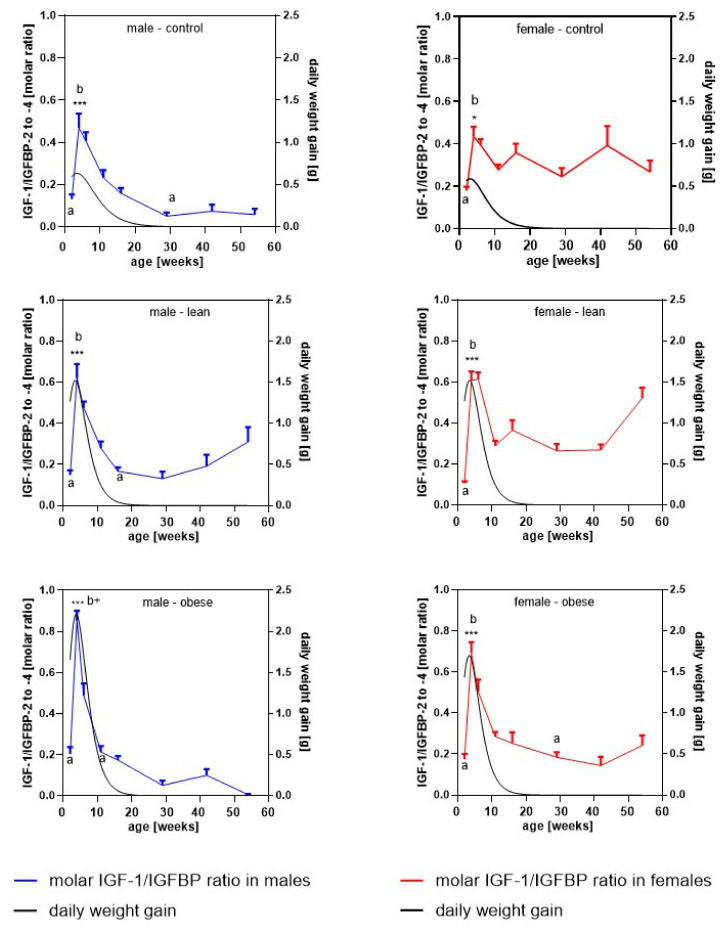
Molar ratio of IGF-1 to the sum of IGFBP-2, -3, and -4 (left Y-axis) present in serum from mice selected for high body weight (obese), for high protein amount (lean), and in unselected controls at an age of 2, 4, 7, 11, 16, 29, 42, and 54 weeks (*n* ≥ 7 with the exception of obese male at 54 weeks with *n* = 4; different letters indicate significant effects of age; identical letters indicate no statistically significant difference; *: *p* < 0.05; ***: *p* < 0.001; b+: significantly different if compared to unselected controls, with *p* < 0.001 for males and *p* < 0.05 for females). On the right Y-axis, daily weight increases, intrapolated from the body weight data in Figure 1 by the Gompertz function, were included.

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
