# Peer review of "Overlap of Peak Growth Activity and Peak IGF-1 to IGFBP Ratio: Delayed Increase of IGFBPs Versus IGF-1 in Serum as a Mechanism to Speed up and down Postnatal Weight Gain in Mice"

_cells, 2020, doi:10.3390/cells9061516_

Round 1

Reviewer 1 Report

In this manuscript, the authors present their findings from studies of IGF-I and IGFBP levels throughout the lifespan of 2 growth enhanced mouse models and control mice. They conclude that early life increases in IGF-I levels are associated with enhanced growth in the 2 models, and that later rises in IGFBP levels may then impair IGF-I bioactivity and decelerate growth. This is an interesting study that provides a provocative in vivo hypothesis for future studies. A number of issues should be addressed.   MAJOR COMMENTS   1. Western ligand blotting is at best semiquantitative, and the only structural information it provides is apparent molecular weight (line 61). More details of the quantitative aspects of the assays should be given e.g. sensitivity and intra- and interassay CVs for each IGFBP.   2. The discussion of IGF and IGFBP levels should acknowledge the inter-relationships between them, e.g. the major determinant of IGFBP-3 levels is GH/IGF-I. Further, the use of the IGF/IGFBP molar ratio is oversimplified since almost all IGFBP-3 is contained within a ternary complex, which has markedly different pharmacokinetic properties than binary complexes. Finally, given the high binding affinity of IGFBPs for IGFs, how much would the changes in molar ratio (between 0.2-0.6) change the amount of free IGF-I? Presumably, almost all would be IGFBP-bound under these conditions?   3. A paragraph on study limitations would be useful. Although not always clarified, the number of mice studied was 4-8/group, which would limit both the power and reproducibility of the findings. Further, the results are purely correlative; suggestions of how the hypothesis could be confirmed would be informative.     MINOR COMMENT   1. Fig 3 (upper): IGF-I levels in obese mice at 54 weeks appear to be 0. Is this correct? How many animals were studied? If only 4 survived to this time point, could there be a survivor effect?

Author Response

Reviewer 1
In this manuscript, the authors present their findings from studies of IGF-I and IGFBP levels throughout the lifespan of 2 growth enhanced mouse models and control mice. They conclude that early life increases in IGF-I levels are associated with enhanced growth in the 2 models, and that later rises in IGFBP levels may then impair IGF-I bioactivity and decelerate growth. This is an interesting study that provides a provocative in vivo hypothesis for future studies. A number of issues should be addressed.

Author response: We want to express our gratitude for the critical and constructive responses. We considered all of the suggestions.

MAJOR COMMENTS 1. Western ligand blotting is at best semiquantitative, and the only structural information it provides is apparent molecular weight (line 61). More details of the quantitative aspects of the assays should be given e.g. sensitivity and intra- and interassay CVs for each IGFBP.
Author response: We provide more details on the method of quantitation for all IGFBPs including lower limits of quantification and intra- and interassay variations in the revised manuscript (lines 106ff). We included a dilution curve of recombinant standards for every single IGFBP with each assay, similar to other quantitative assays. Accordingly the regression coefficients are higher than 0.99 for all IGFBPs. All information is presented online (http://www.ligandis.de/index.php?id=20&L=1).

2. The discussion of IGF and IGFBP levels should acknowledge the inter-relationships between them, e.g. the major determinant of IGFBP-3 levels is GH/IGF-I. Further, the use of the IGF/IGFBP molar ratio is oversimplified since almost all IGFBP-3 is contained within a ternary complex, which has markedly different pharmacokinetic properties than binary complexes. Finally, given the high binding affinity of IGFBPs for IGFs, how much would the changes in molar ratio (between 0.2-0.6) change the amount of free IGF-I? Presumably, almost all would be IGFBP-bound under these conditions?

Author response: We agree with the Reviewer and now acknowledge inter-relationships between the compounds from the IGF-system, explain the simplistic approach provide by the present manuscript, and also ask the question of physiological relevance of molar ratios << 1 (lines 340ff).

3. A paragraph on study limitations would be useful. Although not always clarified, the number of mice studied was 4-8/group, which would limit both the power and reproducibility of the findings. Further, the results are purely correlative; suggestions of how the hypothesis could be confirmed would be informative.

Author response: We included a separate paragraph on the limitations as suggested in the revised manuscript (lines 340ff).

MINOR COMMENT 1. Fig 3 (upper): IGF-I levels in obese mice at 54 weeks appear to be 0. Is this correct? How many animals were studied? If only 4 survived to this time point, could there be a survivor effect?

Author response: We confirm that IGF-1 levels in obese mice at an age of 54 weeks are almost zero: The concentrations of IGF-1 in serum from the four obese male mice at an age of 54 was: 0.18, 6.6, 22.1, and 5.2 ng/ml. In unselected male controls, IGF-1 concentrations were <10 ng/ml in three from a total of 8 mice at an age of 54 weeks. Obese mice have a dramatically reduced life-span (≈50% of unselected controls). Thus, 54 weeks are behind normal life span, and it is difficult to obtain obese male mice at all for this age group. We have a separate manuscript on this fact and mention accordingly in the revised manuscript. We have explained the reduced number in male obese mice in lines 85ff.

Reviewer 2 Report

The paper from Walz and Colleagues investigates the role of IGF-I and BP in post-natal weight gain in different mice models of both genders.

Line 28: the authors state the absence of an increase of IGFBPs in serum but figures 5 and 6 show an increase in IGFBP-3 and 4 in obese and lean mice vs control between weeks 2 and 11. Please explain this discrepancy.

Figure 1: letters indicating different p values are not specified. X-axis: days or weeks?

Figure 2: The Authors used different axis scales between male and female mice. Moreover, it should be better to focus the attention between weeks 0 and 30 to underline the difference in weight gain

The paragraph “3.2. The effects of growth selection on the concentrations of IGF-1” is confused and difficult to understand.

Line 136: remove “concentration of”

Line 138/139: the authors affirm that the increase of IGF1 is gender independent only in control mice. However, figure 3 suggests that gender does not influence the increase also in the obese and lean group

Line 139: the phrase is not in contrast to the previous one

Line 144: Figure 3 seems to show a significant difference between lean and control mice in the all-time point tested. Why the authors focus the attention only on the age of 7 and 54 weeks?

Figure 3: the figure does not present all the p values indicated. It is better to modify the X scale to facilitate the identification of the time points described.

Line 154/157: the reference to the figure is missing

The paragraph “Effects of age and gender on levels of IGFBP2- to 4 in serum from control mice” is confused.

Figure 4: the figure legend is not complete, the p values are not well described and not reported in the figure. The red line for females is missing in the legend. Moreover, this figure does not add any crucial information.

The paragraph “Effects of growth selection on the concentrations of IGFBP-2 to -4” does not report all di differences visible in fig. 5/6 among different genotypes

Line 181: It is not clear if the authors are talking about different serum concentration during the time or between obese and control mice

Line 182: what about weeks 16, 29 for IGFBP-3 concentration vs control?

Figure 8: express X-axis in weeks, check the figure legend

Line 241: remove “if compared”

Line 243: data already published (an increase of IGF-1 serum levels in DU6 and DU6P mice – Timtchenko et al 1999)

Line 254/255: the phrase is not clear

Author Response

Reviewer 2

The paper from Walz and Colleagues investigates the role of IGF-I and BP in post-natal weight gain in different mice models of both genders.
Author response: We want to express our gratitude for the careful review of our manuscript. We addressed all questions and corrected all errors according to the Reviewers suggestions.
Line 28: the authors state the absence of an increase of IGFBPs in serum but figures 5 and 6 show an increase in IGFBP-3 and 4 in obese and lean mice vs control between weeks 2 and 11. Please explain this discrepancy.

Author response: Between 2 and 11 weeks of age a significant increase was found for IGF-1 but not for IGFBPs. We removed the unclear statement in the revised manuscript (lines 26ff).

Figure 1: letters indicating different p values are not specified. X-axis: days or weeks?

Author response: Specification has been performed in the revised manuscript (***: p<0.001; line 143). We corrected X-axis for weeks (line 139).

Figure 2: The Authors used different axis scales between male and female mice. Moreover, it should be better to focus the attention between weeks 0 and 30 to underline the difference in weight gain

Author response: Different Y-axis scales for males and females are used almost for all graphs. This is done in order to better present the data. If we focus only on weeks 0 to 30 we would break the rule of full display followed by all other Figures. In addition the reader might ask why we only provide half of the data for Figure 2. In fact it might be that body weight is increased also at later timepoints e.g. in obese mice. Therefore, we would like not to remove the second half of life in Figure 2. Please keep in mind that also Figure 7 covers the full lifespan.

The paragraph “3.2. The effects of growth selection on the concentrations of IGF-1” is confused and difficult to understand.

Author response: We revised paragraph 3.2. in order to improve clarity (lines 151ff).

Line 136: remove “concentration of”

Author response: This error was removed with the revision of paragraph 3.2. (lines 151ff).

Line 138/139: the authors affirm that the increase of IGF1 is gender independent only in control mice. However, figure 3 suggests that gender does not influence the increase also in the obese and lean group

Author response: We wanted to express by this statement, that an increase in controls was only found if males and females are combined. With the revision clarity is also improved with this issue (lines 151ff).

Line 139: the phrase is not in contrast to the previous one

Author response: This unclear expression is likewise solved in the revised manuscript (lines 154ff).

Line 144: Figure 3 seems to show a significant difference between lean and control mice in the all-time point tested. Why the authors focus the attention only on the age of 7 and 54 weeks?

Author response: We revised the results section accordingly: As a main effect of growth selection independent of age, IGF-1 was significantly increased in lean mice (p<0.001) of both genders but only in obese male mice (p<0.01) if compared to sex-matched unselected controls (Figure 3). Lines 152ff.

Figure 3: the figure does not present all the p values indicated. It is better to modify the X scale to facilitate the identification of the time points described.
Author response: All p values are now included (line 162ff). We harmonized the X-scale of figure 3 with all other Figures, in order facilitate identification of the time points.

Line 154/157: the reference to the figure is missing

Author response: Figure 4 has been deleted from the original manuscript because the content is presented in figures 5 and 6. This is a specific suggestion also from Reviewer 3.

The paragraph “Effects of age and gender on levels of IGFBP2- to 4 in serum from control mice” is confused.

Author response: This part was completely revised and structured in order to generate clarity (lines 168ff).

Figure 4: the figure legend is not complete, the p values are not well described and not reported in the figure. The red line for females is missing in the legend. Moreover, this figure does not add any crucial information.

Author response: This recommendation is also in line with Reviewer 3. Therefore, we removed former figure 4 and related content as described above.

The paragraph “Effects of growth selection on the concentrations of IGFBP-2 to -4” does not report all di differences visible in fig. 5/6 among different genotypes

Author response: The statistical analysis revealed hundreds of significant differences. It is impossible to mention all of them in the results section. Instead we have tried to identify the key features of regulation in the results section. In addition we laid a particular focus on the hypothesis of the manuscript. We generally reconstructed also this part of the description and hope having improved clarity (lines 203ff).

Line 181: It is not clear if the authors are talking about different serum concentration during the time or between obese and control mice
Author response: In line 181 we mentioned the gender effect on IGFBP-4. In the revised manuscript, we more extensive explain, which statistical model is considered, respectively. Thereby, it is more clear now if an effect of age, gender, or mouse line is considered. The content of line 182 was is now present in a revised form in lines 169ff. Here the content of former Figure 4 (direct comparison of IGFBP-2 to -4 in males versus females) is expressed in words.

Line 182: what about weeks 16, 29 for IGFBP-3 concentration vs control?

Author response: Significantly different concentrations of IGFBP-3 are only found at an age of 11 and 16 weeks in obese or lean mice if compared to controls (lines 209ff). Only in lean mice IGFBP-3 remains elevated at weeks 29 and 54 if compared to controls as described in the results section (lines 211ff).

Figure 8: express X-axis in weeks, check the figure legend

Author response: Former figure 8 (now Figure 7) has been modified according to the Reviewers suggestion (line 241).

Line 241: remove “if compared”

Author response: The error has been eliminated (line 269)
Line 243: data already published (an increase of IGF-1 serum levels in DU6 and DU6P mice – Timtchenko et al 1999)

Author response: We have cited appropriately (line 271).

Line 254/255: the phrase is not clear

Author response: We have rephrased: Although daily weight gain declines between 17 and 27.5 days of age in all mice included in that study, the concentrations of IGF-1 in the circulation are remained elevated until an age of at least 7 weeks The elevated levels of IGF-1 thereby cannot explain the massive reductions of growth activity in mice between 4 and 7 weeks of age (lines 277ff).

Reviewer 3 Report

Major comments

1. The authors must write how many mice used in each group in Materials and Methods section. Moreover, they must write why the number of mice in each group is different, 4, 7, and 8. Why the most important male obese group was finally 4 ? Four mice died of obese induced diseases ?

2. The authors should use “weeks” but nor “days” in X-axes and should use marks in the same time points in all figures. They should explain all words in figures. What is “a, b, c, d” in Fig.1 ? What is “a and b” in Figs 3, 5, and 6 ?  What is “a” in Fig.8 ?

3. Figure 4 is unnecessary as it is constructed with several parts of Figures 5 and 6.

4. Figure 7 is unnecessary as it is mixtures of Figure 3 and Figures 5 or 6. 

5. They should analyze molar ratios of IGF1/IGFBP2, IGF1/IGFBP3, and IGF1/IGFBP4, separately.

6. The authors wrote several data with comparison, however they did not write those subjects. e.g. lines 136 & 137, “In obese and lean mice of both genders, serum concentrations of IGF-1 significantly (p<0.001) increased between 2 and 7 weeks of age (Figure 3). “ line 151 & 152, “the highest levels of IGFBP-2 in serum from female control mice were observed at the age of 11 weeks (Figure 4) with significant reductions at week 16 (p<0.01) and week 54 (p<0.05).”

7. The authors must write which post-hoc test used for multiple groups in ANOVA. 

8. The manuscript is not matured. There are several problems. e.g. line 136 “the concentrations of serum concentrations of IGF-1”.

Author Response

Reviewer 3
The authors must write how many mice used in each group in Materials and Methods section. Moreover, they must write why the number of mice in each group is different, 4, 7, and 8. Why the most important male obese group was finally 4 ? Four mice died of obese induced diseases ?

Author response: Male obese mice have a severely reduced life-expectancy if compared to unselected controls and normally do not reach an age of 1 year. Accordingly it was difficult to have healthy obese mice at an age of 54 weeks even if we started with higher numbers at the beginnings. We have provided this information in the revised manuscript. Please note that we consider the younger mice characterized by high daily weight gains as being more important than the eldest and the paper deals more with “peak growth activity” but less with senescence. For the hypothesis the most important groups are the younger groups (week 2 to 16). In the revised manuscript we have provided information on the sample number and we explained why the expected full set of 8 samples per group could not be achieved for all groups (line 84ff and 94ff).

2. The authors should use “weeks” but nor “days” in X-axes and should use marks in the same time points in all figures. They should explain all words in figures. What is “a, b, c, d” in Fig.1 ? What is “a and b” in Figs 3, 5, and 6 ? What is “a” in Fig.8 ?

Author response: We have solved these inconsistencies in the revised manuscript (Figure 1 and 7). We checked all words in figures and revised all figure captions (lines 141ff, 146ff. 163ff, 186ff, 222ff, 227, and 241ff) Within each figures, different letters indicate significant differences. Accordingly “a” versus “b” indicates statistical significance. In order to further specific the level of significance we have added one to three “*” which indicate p<0.05, p<0.01, or p<0.001. Identical letters indicate no statistically significant difference, this information was also included were appropriate (captions for Figures 1, 3, 4, 5, and 7).

3. Figure 4 is unnecessary as it is constructed with several parts of Figures 5 and 6.

Author response: Figure 4 was removed from the revised manuscript.

4. Figure 7 is unnecessary as it is mixtures of Figure 3 and Figures 5 or 6.

Author response: We can understand the Reviewers point. However, we really were surprised about the clarity of delayed rise of IGFBP-3 versus IGF-1 in serum. I do not know any example, which more convincing could demonstrate this novel finding. Therefore we would like to leave Figure 7 included with the manuscript and kindly ask Reviewer 3 for his consent.

5. They should analyze molar ratios of IGF1/IGFBP2, IGF1/IGFBP3, and IGF1/IGFBP4, separately.

Author response: According to the Reviewer´s suggestion, we analyzed all molar ratios separately. Due to very low concentrations of selected IGFBPs in a number of serum samples particularly from elder mice, the molar ratios of IGF-1 to selected IGFBPs ranged between 0.1 and almost 70. If samples with IGFBP-concentrations from 10 to zero were excluded the sample numbers were unbalanced. We therefore chose a method of only presenting groups represented by at least 6 samples. We come to the conclusion that presentation of ratios of IGF-1 to single IGFBPs has several disadvantages: 1) the ratios of IGF-1 versus single IGFBPs are higher than 1 also during adulthood, this may imply more free than bound IGF-1 (which is not the case); 2) acute regulation of IGF-related bioactivity during postnatal peak growth is less clear (significant increases between week 2 and week 4 is only present in4 of 6 groups); 3) the concerted control of IGF-related bioactivity is less well provided. Because the integration of major IGFBPs detected by Western ligand blotting provides an idea on the joint regulation of IGF-related bioactivity during longitudinal growth, we would prefer to keep the original Figure 7 in the revised manuscript. We have provided the new results as supplementary material for the review process.

6. The authors wrote several data with comparison, however they did not write those subjects. e.g. lines 136 & 137, “In obese and lean mice of both genders, serum concentrations of IGF-1 significantly (p<0.001) increased between 2 and 7 weeks of age (Figure 3). “ line 151 & 152, “the highest levels of IGFBP-2 in serum from female control mice were observed at the age of 11 weeks (Figure 4) with significant reductions at week 16 (p<0.01) and week 54 (p<0.05).”

Author response: In the revised manuscript the subjects are revised by a strict inclusion of the statistical model making clear the levels of comparisons (lines 131, 152ff, 169ff, 176ff, 191ff, 198 ff, 204ff, 209ff, 237, 240).

7. The authors must write which post-hoc test used for multiple groups in ANOVA.

Author response: We performed multiple pairwise comparisons which implies that we do not need a preceding F-test. The procedure for multiple comparisons was given in the statistics section: “Least square means (LS means) and their standard errors (SE) were computed for each fixed effect in the models described above, and all pairwise differences between LS means were tested using the Tukey-Kramer procedure.”( lines 121ff)
Please find a detailed response from Dr. Tuchscherer, who performed the statistical analysis: Multiple pairwise comparisons (e.g. Tukey or Tukey-Kramer) control the maximum experimentwise error rate (MEER) for LS-means comparisons, and therefore do not require a preceding F test (Ryan, 1959). A preliminary F test controls the experimentwise error rate under the complete null hypothesis (EERC) but not the MEER. Games (1971) demonstrated that the F test may not be completely consistent with the results of a pairwise comparison approach. Consider, for example, a researcher who is instructed to conduct Tukey's test only if an alpha-level F test rejects the complete null. It is possible for the complete null to be rejected but for the widest ranging means not to differ significantly. This is an example of what has been referred to as incoherence (Gabriel, 1969) or incompatibility (Lehmann, 1957). On the other hand, most multiple-comparison methods (e.g. Tukey-Kramer procedure) can find significant contrasts when the overall F test is nonsignificant and, therefore, suffer a loss of power when used with a preliminary F test (SAS Institute Inc. 2009. SAS/STAT ® 9.2 User’s Guide, Second Edition. Cary, NC: SAS Institute Inc.; p. 2518).
References: Gabriel, K.R., 1969. Simultaneous test procedures – Some theory of multiple comparisons. Annals of Mathematical Statistics 40, 224-240. Hayter, A.J., 1984. A Proof of the Conjecture That the Tukey-Kramer Method Is Conservative. The Annals of Statistics 12, 61–75. Hayter, A.J., 1989. Pairwise Comparisons of Generally Correlated Means. Journal of the American Statistical Association 84, 208–213. Kramer, C.Y., 1956. Extension of Multiple Range Tests to Group Means with Unequal Numbers of Replications. Biometrics 12, 307–310. Lehmann, E.L., 1957. A theory of some multiple decision-problems, 1. Annals of Mathematical Statistics 28, 1-25. Lehmann, E.L., 1957. A theory of some multiple decision-problems, 2. Annals of Mathematical Statistics 28, 547-572. Ryan, T.A., 1959. Multiple Comparisons in Psychological Research. Psychological Bulletin 56, 26-47. SAS Institute Inc. 2009. SAS/STAT ® 9.2 User’s Guide, Second Edition. Cary, NC: SAS Institute Inc.; “Multiple Comparisons.” p. 2514-2524. Tukey, J.W., 1952. Allowances for Various Types of Error Rates. Unpublished invited address presented at Blacksburg meeting of Institute of Mathematical Studies. Tukey, J.W., 1953. The Problem of Multiple Comparisons. In H. I. Braun, ed., The Collected Works of John W. Tukey, volume 8, 1994, New York: Chapman & Hall.

8. The manuscript is not matured. There are several problems. e.g. line 136 “the concentrations of serum concentrations of IGF-1”.

Author response: We have substantially improved clarity and eliminated errors in the revised manuscript and also corrected the problem originally located in line 136 (lines 150ff).
We want to express our gratitude for the critical revision of our manuscript.

Round 2

Reviewer 3 Report

Major comments

  1. Their method of this murine experiment is not appropriate, as they stated male obese mice are difficult survive at age of 54 weeks. Thus the data of 54 weeks are not reliable. The authors should write why the number of some groups are 7, but not 8.?

  1. What is “a, b, c” in Figures ? They wrote different letters indicate significant differences, however the sentence can not be understand at all. Why they use same “letter, a” in the different time points in each figure ? Obese mice were compared only to control ? or to lean ? Lean mice were compared to control only ????

  1. It is unfair to make multiple figures using same data. Thus, Figure 6 is unnecessary as it is mixtures of Figure 3 and IGFBP3 in Figures 4 or 5. If the authors want to show Figure 6, they should delete Figure 3 and IGFBP3 in Figures 4 or 5.

  1. Figure 7 contains all data of Figure 2. Thus, Figure 2 should be deleted. The authors should use same scale (0.0 - 2.5) of daily weight gain in each panel in Figure 7. 

Author Response

With this second major revision, the authors addressed all issues raised by the reviewer and answered all questions. In order to visualize all chances with respect to the review process all changes are labeled in blue (changes by he 1st revision in yellow).

Major comment 1: 

Their method of this murine experiment is not appropriate, as they stated male obese mice are difficult survive at age of 54 weeks. Thus the data of 54 weeks are not reliable. The authors should write why the number of some groups are 7, but not 8.?

Author response:

 In the revised manuscript in the discussion section “limitations” it is stated that the data with n=4 is not reliable. In addition we explained why in some groups one data point is missing. After the first “major revision” we had stated: “Using the software GraphPad Prism version 8.4.2, three samples were identified as outliers (GraphPad Prism) and therefore excluded from further analysis. Accordingly, in male controls at an age of 2 weeks, in obese males at an age of 42 weeks, and in female controls at an age of 42 only 7 samples per group were included (N=377).” In the second “major revision we have marked this statement in blue.

 Major comment 2:

What is “a, b, c” in Figures ? They wrote different letters indicate significant differences, however the sentence can not be understand at all. Why they use same “letter, a” in the different time points in each figure ? Obese mice were compared only to control ? or to lean ? Lean mice were compared to control only ????

Author response:

Different letters indicate significant effect within group (if there are letters present) and between different groups if labeled by letters. Accordingly, “b” in obese mice is different to an “a” at different ages in the same group but also to an “a” of other groups. This is a particular strength of the method. This method is particularly useful when multiple comparisons have to be concerned. The method of using different superscripts for indicating significant differences is used by multiple publications and by many groups. As one of many other examples please e.g. see doi: 10.1136/gut.2008.165886.

I have included more information to indicate the information content of the labeling system in the revised manuscript.

Figure 1 upper panel: At an age of 19 weeks all groups have different weights (a vs b; a vs c and b vs c). The figure thus shows that body weights are different also in obese vs lean mice. In females this is not the case: obese or lean mice are not different between each others (thus they have the same letter) but both groups are heavier than controls. Thus obese and lean females have letters (b) different from controls (a).

Figure 2: Significant increases from the 2 weeks of age (as) to later time points (b) and significant decreases tot he end of life (a). At the end of life the levels are no more different from the beginning (both groups have „a“).

Figures 3 to 5: Significant increases with age (a versus b) but no difference between groups (all groups with an „a“ are not different and all groups with a „b“ are not different).

 Major comment 3:

It is unfair to make multiple figures using same data. Thus, Figure 6 is unnecessary as it is mixtures of Figure 3 and IGFBP3 in Figures 4 or 5. If the authors want to show Figure 6, they should delete Figure 3 and IGFBP3 in Figures 4 or 5.

Author response:

Figure 6 was removed according to the Reviewers suggestions.

 Major comment 4:

Figure 7 contains all data of Figure 2. Thus, Figure 2 should be deleted. The authors should use same scale (0.0 - 2.5) of daily weight gain in each panel in Figure 7. 

Author response:

Figure 2 was removed and the scale of former Figure 7, now Figure 5, was harmonized according to the Reviewers suggestions.